**Subject Category:**
Biology (whole organism)

palaeontology

Varanopidae, Mesenosaurus, Permian, Synapsida

**Author for correspondence:**
Robert R. Reisz
e-mail: robert.reisz@utoronto.ca

# A new varanopid synapsid from the early Permian of Oklahoma and the evolutionary stasis in this clade

## Sigi Maho[1], Bryan M. Gee[1] and Robert R. Reisz[2]

[1]Department of Biology, University of Toronto at Mississauga, 3359 Mississauga Road, Mississauga, Ontario, Canada L5L 1C6
[2]International Centre of Future Science, Jilin University, 2699 Qianjin Avenue, Changchun, Jilin Province, People's Republic of China

  SM, 0000-0002-4983-2655; BMG, 0000-0003-4517-3290; RRR, 0000-0002-7454-1649

Varanopids are a basal clade of small- to medium-sized non-therapsid synapsids, whose range extends from the late Pennsylvanian to the late middle Permian, and are found in North America, Russia, Europe and South Africa. The greatest varanopid diversity is observed at the fossiliferous cave deposits near Richards Spur, Oklahoma, well known for the preservation of a complex early Permian upland community. Two previously described varanopids, *Mycterosaurus* and *Varanops*, are known only from fragmentary disarticulated material at Richards Spur. A third putative varanopid, *Basicranodon fortsillensis*, represented by a partial parasphenoid, has been synonymized with *Mycterosaurus longiceps*. This study reports on a new varanopid taxon, represented by substantially more complete material, including three nearly complete skulls. Such comprehensive cranial material allows for a detailed study of the taxon and its relationship to other varanopids. This new varanopid bears great morphological similarity to *Mesenosaurus romeri* from the middle Permian Mezen River Basin of northern Russia. Phylogenetic analysis recovers a sister relationship between this taxon and *Me. romeri*. This relationship, in conjunction with a detailed morphological comparison, supports the placement of this taxon within *Mesenosaurus*, as a new species, *Me. efremovi*. These results reveal an unexpected extension of the geographical and temporal range of *Mesenosaurus*, contributing to our understanding of varanopid dispersal. The extended persistence of this basal clade of predatory synapsids, underscored by the apparent evolutionary stasis of this genus, is unusual among Palaeozoic tetrapods.

This phenomenon implies an exceptionally high degree of extended ecological resilience across major faunal and environmental transitions.

# 1. Introduction

Varanopids are a clade of small- to medium-sized basal synapsids, with a fossil record in late Pennsylvanian to early Permian localities in North America and Europe that extends into late middle Permian localities in Russia and South Africa [1,2]. This extended fossil record makes them the most widely dispersed group of synapsid amniotes [3] and one of only two basal synapsid clades that extend beyond Olson's gap [4]. Despite these extensive temporal and geographical ranges, varanopids are generally rare in Palaeozoic terrestrial vertebrate assemblages, being typically represented by a single taxon per locality (e.g. [5]). An exception to this is the Richards Spur locality, a series of infilled karst fissures in the Ordovician Arbuckle limestone in Oklahoma [6]. The locality is one of the most productive sites for early Permian tetrapod fossils and captures an upland ecosystem [6]. The assemblage includes representatives of three clades of non-therapsid synapsids: Caseidae [7,8], Sphenacodontidae [9,10] and Varanopidae [5,11]. Recent collection efforts have revealed that varanopids, represented by two taxa that were first described from the Texas red beds, *Mycterosaurus* and *Varanops*, are the most common clade. This study reports on a new varanopid species from this fossiliferous locality that bears great morphological similarity to *Mesenosaurus romeri* from the middle Permian Mezen River Basin in northern Russia [12].

*Institutional abbreviations*—BP, Bernard Price Institute for Palaeontology, University of the Witwatersrand, Johannesburg, South Africa; OMNH, Sam Noble Oklahoma Museum of Natural History, University of Oklahoma, Norman, OK, USA; PIN, Paleontological Institute, Russian Academy of Sciences, Moscow, Russia.

*Anatomical abbreviations*—an, angular; ar, articular; at, atlas; ax, axis; bo, basioccipital; co, coronoid; cp, cultriform process; d, dentary; ec, ectopterygoid; ep, epipterygoid; eo, exoccipital; f, frontal; j, jugal; l, lacrimal; m, maxilla; n, nasal; nd, nasolacrimal duct; op, opisthotic; p, parietal; pal, palatine; pf, postfrontal; pm, premaxilla; po, postorbital; pp, postparietal; pra, prearticular; prf, prefrontal; pro, prootic; ps, parasphenoid; pt, pterygoid; q, quadrate; qj, quadratojugal; s, stapes; sc, sclerotic plates; sm, septomaxilla; so, supraoccipital; sp, splenial; sq, squamosal; st, supratemporal; su, surangular; t, tabular; v, vertebra; vo, vomer.

# 2. Material and methods

Synapsida Osborn, 1903
Eupelycosauria Kemp, 1982
Varanopidae Romer and Price, 1940
Mycterosaurinae Reisz and Berman, 2001
*Mesenosaurus* Efremov, 1938
*Mesenosaurus efremovi* sp. nov.

*LSID*—urn:lsid:zoobank.org:pub:F80E30F1-0425-4A5D-804E-45BFF542D86D

*Diagnosis*—Distinguished from the genotype by the presence of short dorsal premaxillary processes that do not extend to the level of the posterior narial margin and by the greater separation of the premaxillae posteriorly by the nasals; more posteriorly extensive maxilla; fewer maxillary tooth positions; and the presence of a contact between the postorbital and supratemporal bones.

*Etymology*—The specific designation honours the famous Russian palaeontologist, Ivan Efremov, who erected the genus, and is recognized as one of the leaders of the field in his country.

*Holotype*—OMNH 73209, a nearly complete skull and mandible (figures 1 and 2).

*Referred Specimens*—OMNH 73500, a nearly complete skull and mandibles (figures 3 and 4); OMNH 73208, a nearly complete skull and left mandible (figure 5).

*Materials*—The material used in the current study consists of three new specimens assigned to a new species of *Mesenosaurus* (OMNH 73208, OMNH 73209, OMNH 73500). OMNH 73208 consists of a nearly complete skull, dorsoventrally compressed and splayed, exposed in dorsal view. Most of the skull roof, disarticulated posteriorly, partial disarticulated braincase, articulated left mandible and sclerotic rings are preserved. OMNH 73209 consists of a laterally compressed, nearly complete skull that is disarticulated

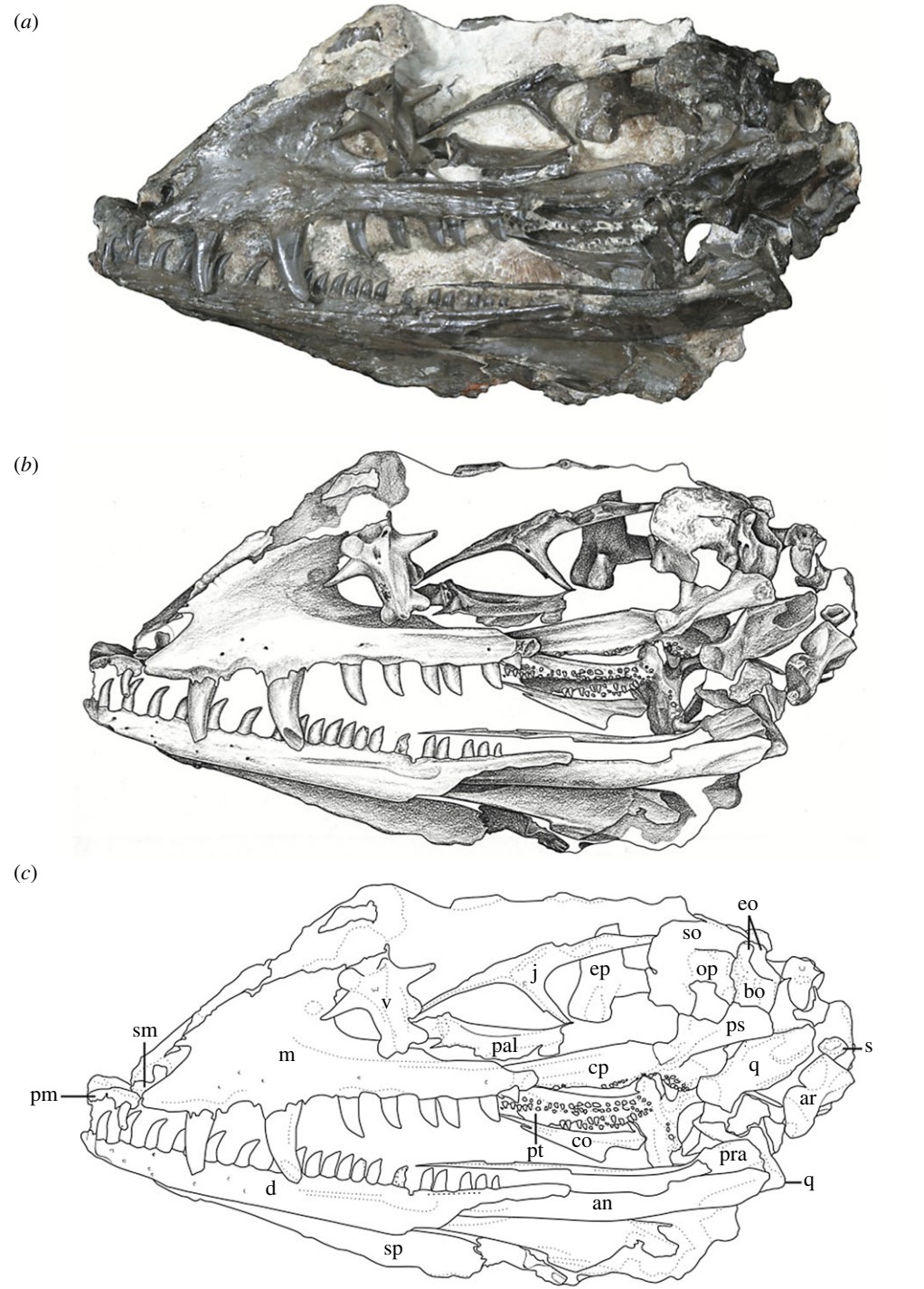

**Figure 1.** *Mesenosaurus efremovi*, sp. nov., holotype, OMNH 73209, nearly complete skull and mandibles in the left lateral view. (*a*) Photograph; (*b*) illustration; (*c*) labelled line drawing. Scale bar, 2 cm.

posteriorly, partial mandibles and a complete but disarticulated braincase. Only small portions of the partially disarticulated left palatal elements are preserved in lateral view. OMNH 73500 consists of a nearly complete skull, split along the midline and laterally splayed, complete and articulated right mandible, partial left mandible, disarticulated brain case and partially disarticulated palate, exposed in lateral view. The block also contains the atlas and axis, disarticulated from the skull. Specimen measurements are listed in table 1.

*Locality and Horizon*—Richards Spur locality (Dolese Brothers Limestone Quarry) near Fort Sill, Comanche County, Oklahoma, lower Permian.

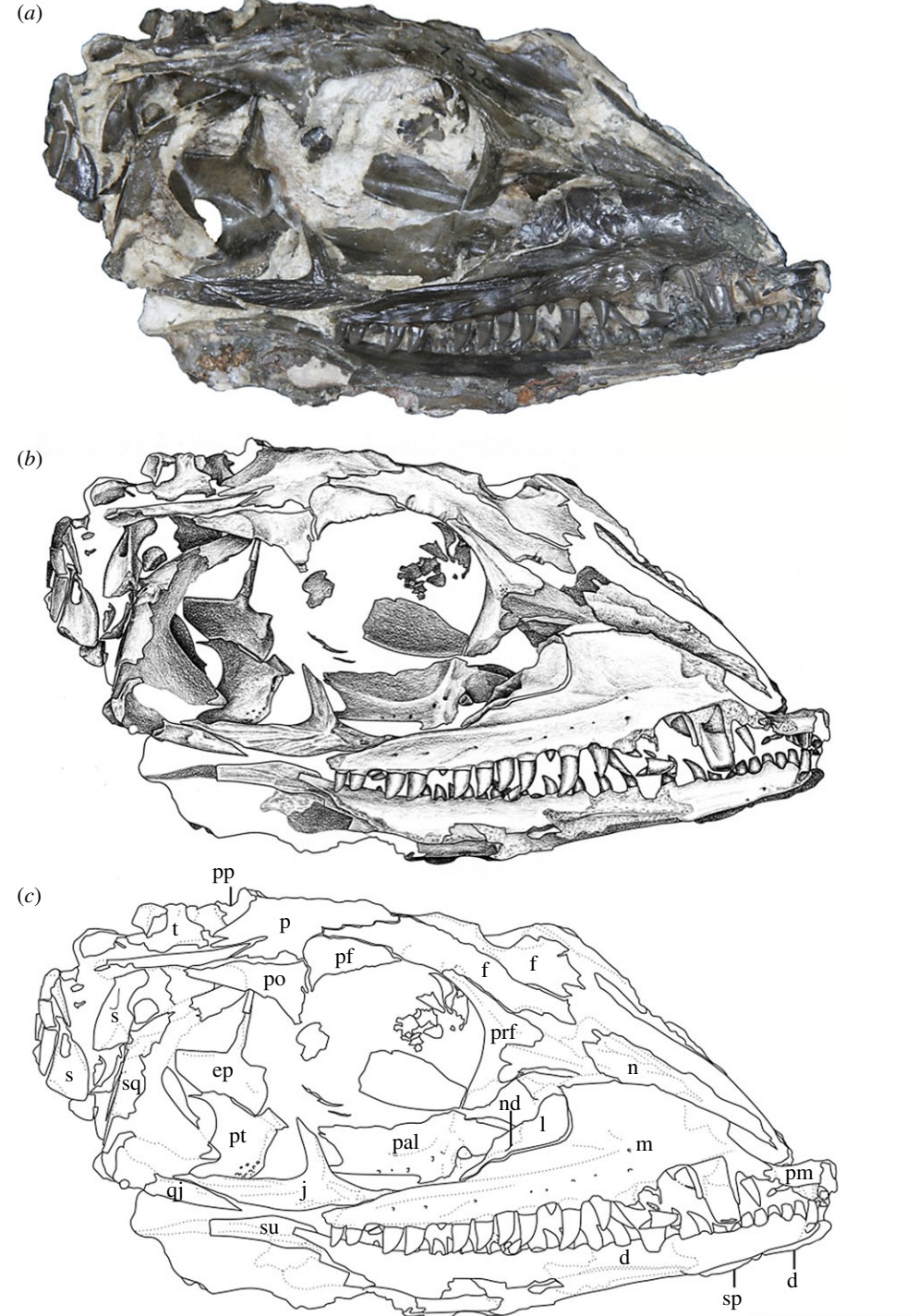

**Figure 2.** *Mesenosaurus efremovi* sp. nov., holotype, OMNH 73209, nearly complete skull and mandibles in the right lateral view. (*a*) Photograph; (*b*) illustration; (*c*) labelled line drawing. Scale bar, 2 cm.

## 2.1. Methods

For this study, we elected to use the matrix of Brocklehurst *et al*. [13] because of its broad taxon and character sampling (58 taxa, 244 characters) compared to more narrowly focused recent studies [14,15]. We elected not to use the analysis of Ford & Benson [14], a derivation of Reisz *et al*. [16], because it is not our goal in this study to further test the hypothesized placement of varanopids within Diapsida. In addition, the inclusion of a clearly derived varanopid with a high degree of anatomical similarity to *Me. romeri* would not be predicted to produce major changes in amniote topology. Similarly, we elected not to use the matrix of Spindler *et al*. [15] because of some

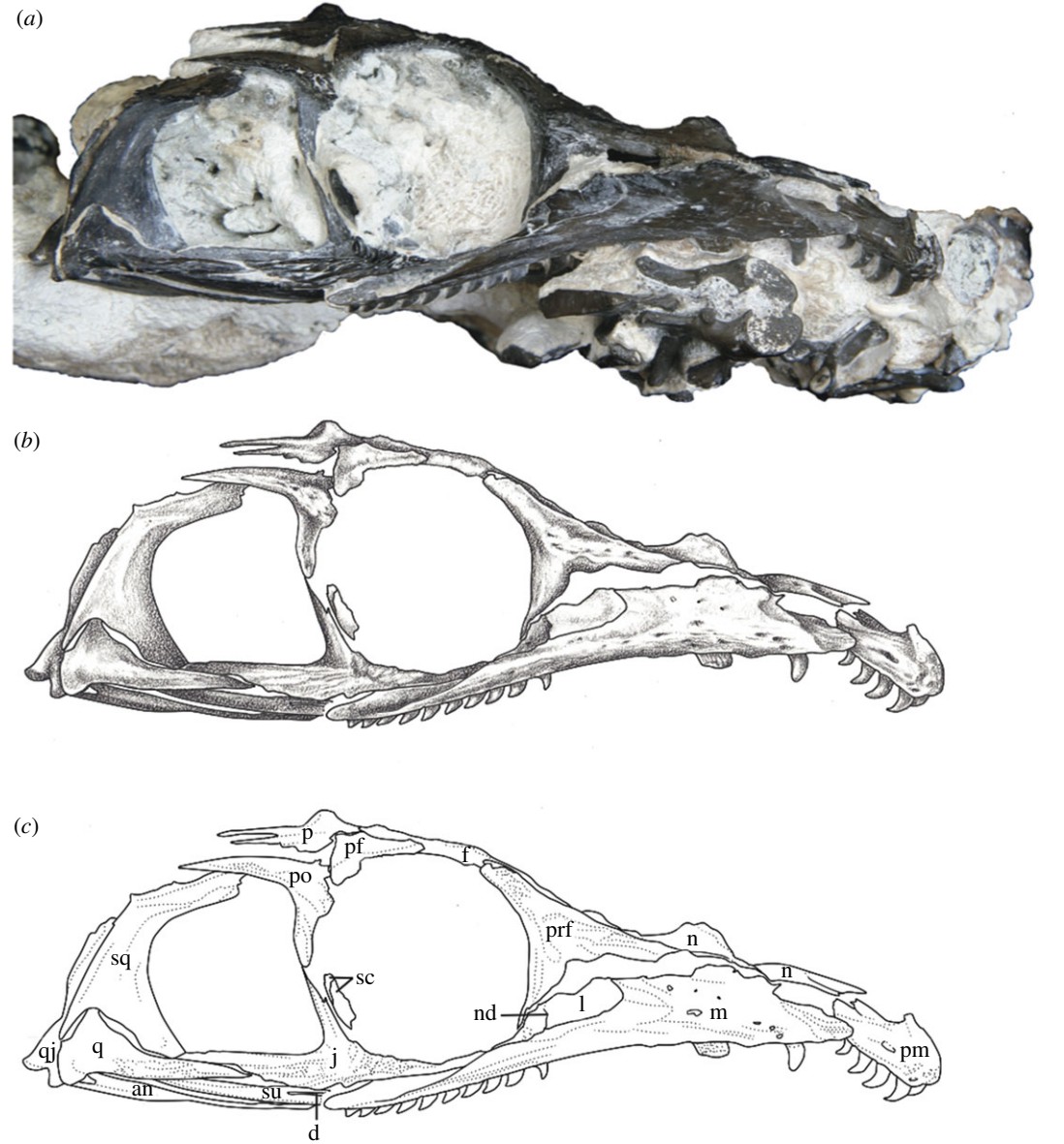

**Figure 3.** *Mesenosaurus efremovi* sp. nov., referred specimen, OMNH 73500, nearly complete skull in the right lateral view. (*a*) Photograph; (*b*) illustration; (*c*) labelled line drawing. Scale bar, 2 cm.

complications that result from the description of *Cabarzia trostheidei*, a taxon known only from a headless skeleton [17]; further comments are provided in Discussion. Character codings were left mostly unchanged from that of Brocklehurst *et al.* [13] except for the following minor corrections. For character 25 (maxilla, ascending process), *Heleosaurus scholtzi*, *Me. romeri*, *Mycterosaurus longiceps* and BP 1 5678 were previously coded for the first derived condition (present, but short and rounded dorsally). We changed these codings to the third derived condition (tall, but also anteroposteriorly long, accommodating deeply implanted tooth roots) because the process is neither short nor dorsally rounded in these taxa, a feature also found in the newly described *Me. efremovi*. We also recommend removing the aspect referring to deep tooth implantation to avoid compound character state construction. For character 34 (caniniform region), *H. scholtzi* and BP 1 5678 were previously coded for the second derived condition (large caniniform tooth, or two teeth distinctly larger than other maxillary teeth). We changed this coding to the first derived condition (caniniform region present) based on the literature; other mycterosaurines were previously coded for this condition. Lastly, the coding for character 74 (jugal–squamosal contact) for *Me. romeri* was previously listed as being absent (74-0); however, figures of Reisz & Berman [12] clearly show a contact (74-1), as in *Me. efremovi*.

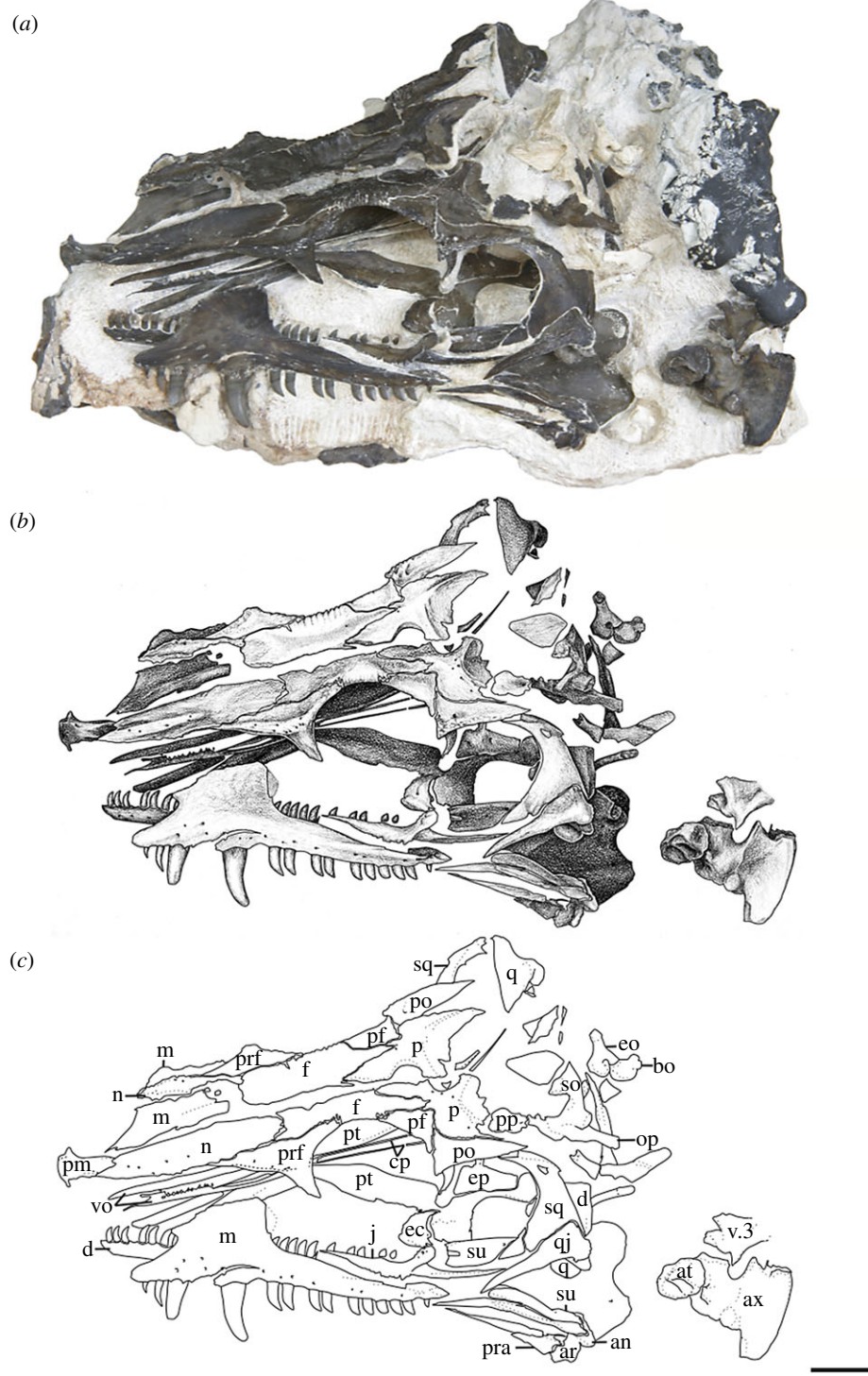

**Figure 4.** *Mesenosaurus efremovi* sp. nov., referred specimen, OMNH 73500, nearly complete skull in the dorsolateral view. (*a*) Photograph; (*b*) illustration; (*c*) labelled line drawing. Scale bar, 2 cm.

## 3. Description

The bones of the skull and mandible range from partial to complete and are generally articulated in OMNH 73209 (holotype), OMNH 73208 and OMNH 73500. The palatal and occipital regions, however, are mainly disarticulated and some approximate positions were deduced from margins of adjacent bones (figures 1–5). The skull is subrectangular in the lateral view, with the posterior margin of the jaw approximately level with the occipital margin and the antorbital region of the skull being longer than the postorbital region (figure 3). The external nares are large and elongate openings

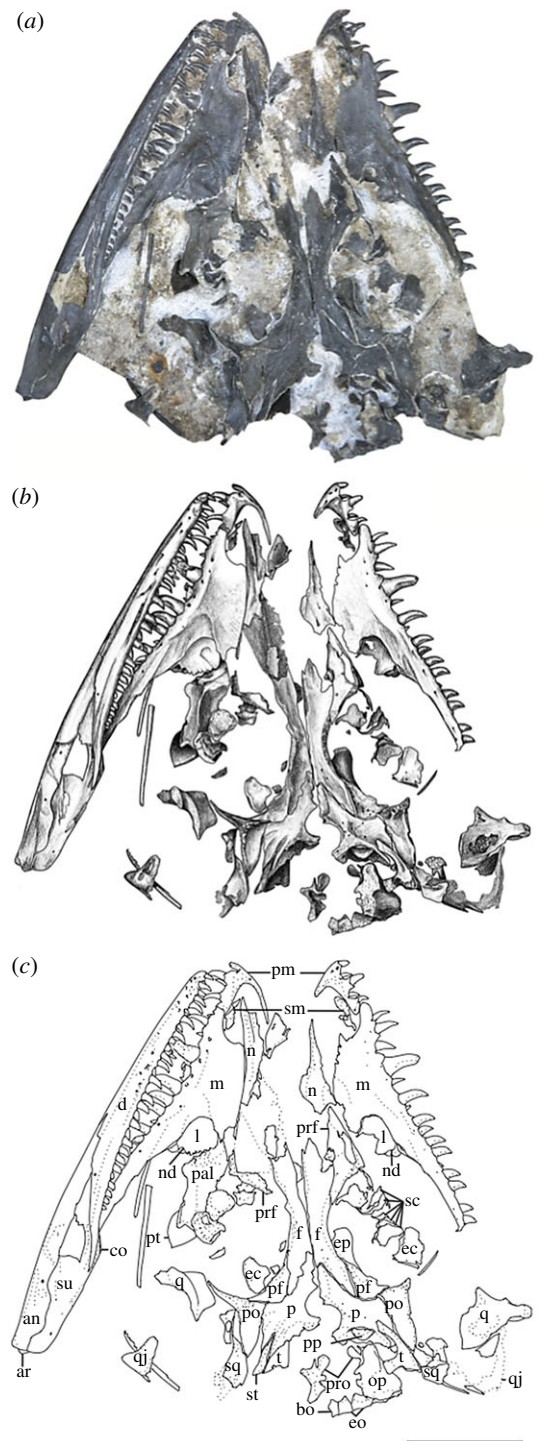

**Figure 5.** *Mesenosaurus efremovi* sp. nov., referred specimen, OMNH 73208, nearly compete skull in the dorsal view. (*a*) Photograph; (*b*) illustration; (*c*) labelled line drawing. Scale bar, 2 cm.

oriented anteromedially and enclosed by the premaxilla, maxilla and nasal. The orbits are large and circular, with an expanded circumorbital rim above the skull roof formed by the prefrontal, frontal and postfrontal. Circumorbital tuberosities are present on the lateral surfaces of the prefrontal, the frontal, the postfrontal, the postorbital and the jugal (figure 4). A large and round pineal foramen is positioned near the posterior margin of the skull roof (figure 4). A large, dorsoventrally elongate, lateral temporal fenestra is present posterior to the orbit, extending close to the ventral edge of the cheek and forming a narrow zygomatic bar (figure 3). Overall, there is little to differentiate *Me. romeri* from *Me. efremovi* and in addition to a basic description of the cranial anatomy of *Me. efremovi*, we will also discuss these differences.

**Table 1.** Measurements of the holotype and referred specimens of *Me. efremovi* and the holotype of *Me. romeri*. All measurements are in centimetres.

| | Mesenosaurus efremovi | | | Mesenosaurus romeri |
|---|---|---|---|---|
| | OMNH 73209[a] | OMNH 73500 | OMNH 73208 | PIN 158/1[a] |
| maximum length | 7.5 | 8 | 6.5 | 4.8 |
| interorbital width | 1.2 | 1.4 | 0.6 | 0.8 |
| orbit height | 2.7 | 2.4 | 1.9 | ? |
| orbit width | 2.1 | 2.2 | 1.8 | 1.8 |
| temporal fenestra height | ? | 2.0 | ? | ? |
| temporal fenestra width | ? | 1.7 | ? | ? |

[a]Holotypes.

## 3.1. Skull roof

The premaxilla is a transversely narrow element that forms the tip of the snout, framing the large external naris anteriorly, dorsally and ventrally (figures 3–5). The anterior margin of the naris is medially recessed, resulting in an expanded narial shelf, smoothly rounded between the medial and lateral surface, and narrowing at the base of the dorsal process (figure 3). The dorsal or nasal process tapers posteriorly on the dorsal surface and partially divides the nasals anteriorly along the midline (figure 4). This process is longer in *Me. romeri* and runs completely along the midline. A ventral subnarial maxillary process contacts the maxilla along the lateral surface at the mid-length of the external naris. The base of the ventral process contributes to the ventral margin of the skull and extends medially on the palatal surface to articulate with the vomer and to form the anterior margin of the internal naris. The premaxilla has space for a maximum of five marginal teeth along the ventral margin, all closely spaced and strongly recurved, with anterior and posterior cutting edges (figure 3). Teeth at the first and second positions appear approximately similar in size, but the third position has a slightly enlarged tooth, the largest of the premaxillary series. This condition is similar to that seen in the genotype, and differs from the condition in *Mycterosaurus*, where the first premaxillary tooth is enlarged. The fourth and fifth teeth are smaller than the third tooth, decreasing posteriorly in size.

The septomaxilla is a laterally concave element positioned within the centre of the narial cavity and preserved in the left external naris of OMNH 73209 (figure 1) and in both nares of OMNH 73208 (figure 5).

The maxilla is the longest bone of the skull, extending along the ventral edge of the skull from the mid-point of the external nares to the temporal region, terminating approximately at the dorsal process of the jugal (figures 1–5). On the more disarticulated left side of OMNH 73500 (figure 4), the suborbital process of the maxilla appears to extend beyond the dorsal process, whereas on the right side (figure 3), the maxilla terminates in line with the dorsal process. Regardless, this element is longer than in *Me. romeri*, where the posterior process of the maxilla does not reach the post-orbital bar. The subnarial process of the maxilla articulates with the premaxilla anteriorly as its posterior surface expands dorsally to frame the posteroventral narial margin. The anteroposteriorly long dorsal process forms the posterior margin of the external nares, elongating posterodorsally to articulate with the nasal and prefrontal dorsally and the lacrimal posteriorly, restricting the lacrimal to the proximity of the anterior orbital margin. The lacrimal extends about 40% the distance between the anterior margin of the orbit and posterior margin of the naris, as in *Me. romeri* [12]. Only the smooth lateral surface of the maxilla is exposed in all three specimens, bearing a series of similarly sized foramina, dorsal to the marginal tooth row, and lateral swelling along the caniniform region (figures 1–5).

The marginal dentition includes space for a maximum of 19 teeth (figures 1–5). Even though the maxilla extends farther posteriorly in *Me. efremovi* than in *Me. romeri*, there are considerably fewer maxillary teeth. We attribute this difference to the comparatively larger teeth present in the new taxon, with three distinct, enlarged teeth being present in *Me. efremovi* in the region of the dorsal process of the maxilla, whereas there are up to seven teeth in the same region in *Me. romeri*. The first tooth is similar in length to the posterior-most tooth of the premaxilla; however, succeeding teeth increase in size, with those in the caniniform region (positions 3–6) being the longest in length and diameter (figures 1, 4 and 5). Posterior to this region, the teeth gradually decrease in size. The

morphology of the teeth is similar to those of the premaxilla except that the maxillary dentition exhibits delicate serrations along the anterior and posterior edges [9].

The lacrimal is a small subtriangular bone with a slight contribution to the anterior margin of the orbit, articulated with the maxilla anteroventrally and prefrontal posterodorsally (figures 1–5). A slender, short posterior process extends along the suborbital margin, medial to the maxilla and appears to contact the jugal. External expression of the nasolacrimal duct is present on the lateral surface, dorsal to the maxilla (figures 2 and 3). The lacrimal is excluded from the external naris by the contact of the dorsal process of the maxilla with the nasal and prefrontal.

The nasal is a subrectangular, elongate element articulated with the premaxilla anteriorly, with the dorsal process of the maxilla ventrally, with the frontal posteriorly and with the prefrontal posterolaterally (figures 1–5). On the dorsal surface, a thin, anteriorly tapering process articulates with the alary process of the premaxilla and forms the posteromedial margin of the external naris (figure 4). Posteriorly, the nasal articulates with the anterior process of the frontal. The nasal exhibits a grooved depression extending anteroposteriorly along its narrow lateral surface just above the dorsal process of the maxilla and extending to the anterior margin of the prefrontal (figure 4).

The prefrontal is a large, subtriangular element contributing to the anterior and anterodorsal orbital margins (figures 2–4). An anteriorly directed process partially divides the posterior end of the maxilla and the nasal. The antorbital process forms the anterior margin of the orbital and articulates with the lacrimal, greatly limiting its contribution to the orbital margin (figure 3). On the dorsal surface, the prefrontal is articulated with the nasal anteromedially and with the frontal posteromedially. The prefrontal is composed of a sharp right-angled transition between the lateral and dorsal surfaces. The bend slightly overhangs the lateral surface of the skull and is accentuated by tubercular ornamentation (figures 2–4).

The subrectangular frontal is the longest midline element of the skull roof in both species of *Mesenosaurus*. The lateral margin contributes to the anterior margin of the orbit, articulating anterolaterally with the prefrontal and posterolaterally with the postfrontal (figures 2–5). The frontal expands anteroposteriorly on the dorsal surface of the skull, and a narrow, posterolateral triangular process tapers between the parietal and postfrontal, terminating at the level of the pineal foramen. Tuberosities are present on the lateral surface of the frontal (figures 2–5).

The postfrontal is a small triangular element and, as in all varanopids, forms the posterodorsal edge of the orbit (figures 2–5). The postfrontal articulates anteromedially with the frontal, posteriorly with the postorbital and posteromedially with the parietal where it is slightly recessed.

The postorbital is a large triradiate element forming the posterior margin of the orbit and the anterodorsal margin of the temporal fenestra (figures 3 and 5). The ventral process of the postorbital articulates with the jugal at approximately the mid-height of the orbit and the temporal fenestra (figure 3). The short anteromedial process of the postorbital wedges between the postfrontal and parietal. The long posterior process articulates with the squamosal along the dorsal margin of the temporal fenestra and with the supratemporal posteromedially. By contrast, in *Me. romeri*, the posterior process of the postorbital does not contact the supratemporal. Ornamentation on the lateral surface of the postorbital forms a protuberance in both species, extending outward past the lateral surface of the skull (figures 3–5).

The parietal is a broad, subtriangular element framing the pineal foramen (figures 4 and 5). The narrow anterior process extends along the midline of the skull and into the supraorbital region, separating the posterolateral processes of the frontals. The wing-like posterolateral process articulates with the medial margin of the postorbital and bears a deep groove (figures 4 and 5). The groove accommodates the mediolaterally thin anterior portion of the supratemporal, only partially preserved in the holotype and OMNH 73208. The large, round pineal foramen is positioned at the posterior half of the parietals, at the level of the posterior orbital margin. The occipital margin of the parietal forms a sloping, dorsally concave posteroventral occipital shelf (figures 4 and 5).

The postparietal is a subrectangular element articulating with the posterior margin of the parietal, meeting its pair at the midline, and overlying the occipital shelf of the parietal (figure 4). The element is only well preserved and identifiable in OMNH 73500, with the medial surface exposed (figure 4).

The tabular is preserved in the holotype and in OMNH 73208 (figures 1, 2 and 5). The convex anterodorsal margin is continuous with the curvature of the parietal occipital margin on either side of the postparietal. As seen in *Me. romeri* and *H. scholtzi*, there is a well-developed medial process distally.

The jugal is a triradiate element contributing to the posteroventral margin of the orbit, the anteroventral margin of the temporal fenestrae and the ventral margin of the skull (figures 1–3). The suborbital process tapers anteriorly to overlap the maxilla, extending slightly along the inner surface of the maxilla (figure 3). The subtemporal process tapers posteriorly to articulate with the quadratojugal, forming a narrow subtemporal bar. The dorsal process articulates with the ventral

process of the postorbital to form a post-orbital bar that frames the orbit and the temporal fenestra. Tubercular ornamentation is present along the suborbital and subtemporal processes (figures 2 and 3).

The squamosal is a sickle-shaped, outwardly convex element forming the posterior margin of the temporal fenestra and the posterior margin of the skull (figures 2–4). The anteroventral process contacts the jugal, excluding the quadratojugal from contributing to the posterior margin of the temporal fenestra (figures 3 and 4). The postorbital process tapers anteriorly to underlie the posterior process of the postorbital. The temporal edge of the squamosal is strongly curved and forms a large temporal fenestra together with the postorbital and jugal. This temporal fenestra appears to be substantially larger than in *Mycterosaurus* and other mycterosaurines, and also appears to be slightly larger than in *Me. romeri*. However, the size of the temporal fenestra may be an ontogenetically variable character, and we avoid incorporating it into the diagnosis of this taxon. There is a posteriorly directed process on the posterodorsal surface that is smaller than that reported in *Elliotsmithia longiceps* [18]. Modesto *et al.* [18] reported the presence of this process in *Me. romeri* based on personal observation and Ivakhnenko *et al.* [19]. Although the feature was not originally reported or figured in the redescription of *Me. romeri* by Reisz & Berman [12], it was personally observed by one of the authors of this study (R.R.R.).

The quadratojugal is an elongate subtriangular element forming the posteroventral corner of the skull roof (figures 3 and 4). The anterior ventral process is slender, extends beyond the level of the anteroventral edge of the squamosal and has an extended sutural contact with the jugal. However, it fails to reach the maxilla (figure 3). The dorsal process of the quadratojugal is overlapped laterally by the squamosal.

## 3.2. Palate

The vomer is an elongate, slender element articulated with the premaxilla anteriorly, the palatine posteriorly and the pterygoid posteromedially (figure 4). The anterior and posterior processes of the vomers are bifurcated asymmetrically with a longer medial spur, similar to *Me. romeri* [12]. Due to dislodgement, the inferred contacts (e.g. anteriorly with the premaxilla) are not readily discernable. Each vomer bears a tooth row on a prominent ridge along the medial edge of the bone (figure 4).

The palatine is a large element that articulates with the vomer anteriorly, the pterygoid medially and the maxilla laterally, and forms the posteromedial portion of the internal naris. The preserved palatines in OMNH 73500 and OMNH 73208 are disarticulated and with only the dorsal surfaces exposed, which bear a central buttress (figures 4 and 5).

The pterygoid is the longest bone of the palate. The pterygoid extends far anteriorly to partially divide the vomers and articulates with the palatine and the ectopterygoid anterolaterally (figure 4). The ventral surface of the pterygoid has three denticulate ridges radiating from the basipterygoid articulation (figure 1). Two sets of teeth are on the palatal ramus of the pterygoid. The first set is a single row oriented parallel to the medial margin that is continuous with the tooth row of the vomer. The second set extends anterolaterally from the basipterygoid articulation and is less tightly spaced than the first series, originating as a single row and expanding into a loose but confined field. Between the ridges of the palatal ramus, the surface is strongly concave. A distinct transverse flange, posterior to the palatal ramus, bears a third ridge with larger teeth arranged in transverse rows along the posterior margin (figure 1). The pterygoid articulates with the epipterygoid dorsally near the basicranial articulation (figures 2 and 4). The slender quadrate ramus of the pterygoid extends posterolaterally, presumably to overlap with the quadrate (figure 4). This ramus is relatively short in height but extends dorsally towards the skull roof, posterior to the epipterygoid (figure 4).

The ectopterygoid is a crescent-shaped posterolateral element articulating with the palatine anteriorly and the pterygoid posteromedially and appears to be edentulous (figure 4).

The quadrate is a robust element composed of a dorsal process and ventral condyles (figures 4 and 5). The dorsal process is anteroposteriorly thin and blade-like, curving anteromedially, and externally sheathed by the quadratojugal and squamosal. The lateral and medial condyles are separated by a deep groove. The quadrate foramen is located at the intersection of the quadratojugal, squamosal and quadrate and is only visible in occipital view.

## 3.3. Braincase

The parasphenoid covers most the ventral surface of the braincase with an anteriorly long and narrow cultriform process and a broad, rectangular basal plate (figure 1). The plate is transversely broad, expanding gradually posterior to the basicranial articulation. Present on the basal plate is a central concave region that is framed by two posterolaterally angled ridges. The anterior junction of these ridges is shallowly depressed (in ventral view) between the basipterygoid processes and merges anteriorly into the cultriform process.

Within the concavity is a very shallowly developed transverse ridge that partially subdivides the depression; this ridge is largely obscured by the dislodged quadrate. Small teeth are present on the anterolateral ridges and extend anteriorly to meet a single row of teeth along the midline of the cultriform process.

The basioccipital articulates dorsally with the exoccipital and forms the ventral portion of the occipital condyle (figures 1 and 4).

The exoccipital is a paired element forming the dorsolateral corners of the occipital condyle and ventrolateral margins of the foramen magnum (figures 1, 4 and 5). They are separated dorsomedially by the supraoccipital and ventromedially by the basioccipital.

The epipterygoid is a palatal element with a subtriangular base ventrally articulated with the quadrate ramus of the pterygoid and a lenticular facet extending anterodorsally to form a dorsally projecting slender columella (figures 2 and 4).

The supraoccipital is a broad, plate-like element of the dorsal occipital surface contributing to the dorsal margin of the foramen magnum and articulating with the exoccipital ventrally and the opisthotic ventrolaterally (figures 1 and 4). A slight median ridge is observed on the dorsal surface.

The opisthotic is fused dorsally with the supraoccipital and forms a ventromedial process contacting the lateral surface of the exoccipital (figures 1 and 4).

The prootic is only well exposed in OMNH 73208 and is largely identified by its position between the parietal and the opisthotic (figure 5). A partially enclosed opening along one margin probably represents the fenestra ovalis.

The stapes comprises a columella and footplate, preserved only in OMNH 73209 but disarticulated (figure 2). The rod-like columella is short and relatively slender. A large stapedial foramen pierces the columella at the junction with the footplate. A small process extending off the footplate (the quadrate process) is partially exposed and appears similar to that reported in *Me. romeri* [12].

## 3.4. Mandible

The dentary is the longest bone in the mandible, extending from the symphysis to the level of the mid-length of the temporal fenestra, forming the anterodorsal margin of the mandible (figures 1, 2 and 5). Posteriorly, the dentary tapers dorsally to a thin process which lies in a shallow groove of the surangular near the coronoid eminence (figure 5). It also meets the angular posteroventrally along the lateral surface. Small foramina are present ventral to the tooth row and concentrated near the mandibular symphysis. There are 31 tooth positions for the strongly recurved teeth, serrated along the anterior and posterior edges and decreasing in size posteriorly (figure 5).

The splenial is an element on the medial and ventral surfaces of the mandible that is only exposed in OMNH 73209, being slightly visible ventral to the dentary (figures 1 and 2).

The coronoid is a triangular element with a narrow labial exposure; preservation obscures the lingual exposure in all articulated specimens. In OMNH 73209, the left coronoid has been disarticulated to show its complete profile, with a tapering anterior process, a dorsoventral expansion posteriorly to frame the coronoid process from within and with a large lingual exposure (figure 1).

The angular is positioned in the posteroventral half of the lateral surface of the mandible, contributing to the ventral margin (figures 1, 3 and 5). It articulates with the surangular dorsally, the dentary anterodorsally and the articular posteriorly.

The surangular is positioned in the posterodorsal half of the lateral surface of the mandible, articulating with the dentary anteriordorsally and the angular ventrally (figures 3 and 5). A gradual dorsal expansion of the surangular forms the slightly convex coronoid eminence observed on the lateral surface.

Only a fragment of the prearticular is preserved in specimen OMNH 73500, situated on the medial surface of the mandible (figure 4). It is a long, slender element.

The articular is positioned on the medial surface of the jaw and can be only partially observed on the lateral surface in OMNH 73500 (figure 4). Extending posteriorly from the angular and surangular, the articular forms a short retroarticular process and the posterior margin of the mandible.

# 4. Discussion

## 4.1. Comparative anatomy

The Richards Spur taxon exhibits numerous varanopid synapomorphies such as a slender subtemporal bar, labiolingually compressed and strongly recurved marginal dentition, and a posteriorly extensive

frontal. Additional features are indicative of mycterosaurine affinities, such as the exclusion of the lacrimal from the external naris and an anteroposteriorly broad dorsal lamina of the maxilla that underlies the nasal and that contacts the prefrontal (figure 3). In addition, the prefrontal has an extensive ventral process that restricts the lacrimal's contribution to the orbit to a narrow sliver (figure 3). More specifically, the taxon is remarkably similar to *Me. romeri* from Russia, sharing features such as lateral swelling of the maxilla in the caniniform region and five premaxillary tooth positions, which have not been reported in other mycterosaurines [12].

There are four morphological differences between *Me. romeri* and *Me. efremovi*: (i) the posterior extent of the dorsal process of the premaxilla; (ii) the posterior extent of the maxillary dentition; (iii) the maxillary tooth count; and (iv) the postorbital–supratemporal contact. The dorsal process of the premaxilla in *Me. efremovi* is relatively short anteroposteriorly. *Mesenosaurus efremovi* has 19 maxillary tooth positions in the tooth row, which terminates ventral to the post-orbital bar (figure 3), whereas *Me. romeri* has 21 positions with the posterior termination anterior to the post-orbital bar [12]. Lastly, the postorbital contacts the supratemporal in *Me. efremovi*, rather than being separated by the parietal (figure 4). Another possible morphological difference between the two taxa is the size of the temporal fenestra, that of *Me. efremovi* being anteroposteriorly longer, producing an opening of more equant proportions relative to *Me. romeri* in which it is more oblong ([12]; figure 4). We did not include this distinction in the diagnosis because the size of the temporal fenestra may be correlated with skull size and thus could be ontogenetically variable. *Mesenosaurus efremovi* is distinctly larger than the largest known specimen of *Me. romeri* (table 1), but it is unclear at present whether they achieved comparable sizes and had similar ontogenetic trajectories. Overall, these differences are not substantial, and we would argue they are insufficient for taxonomic distinction above the species level. We therefore support the inclusion of *Me. efremovi* in *Mesenosaurus* but distinguish it at the species level from the genotype.

One other taxon that merits brief mention is *Basicranodon fortsillensis*, a poorly known varanopid represented by a partial parasphenoid from Richards Spur [20]. Reisz *et al*. [11] noted both the fragmentary nature of the holotype and its indistinguishable nature from *My. longiceps* (at the time, the only mycterosaurine for which the parasphenoidal dentition was known). They presented two options for the status of *B. fortsillensis*: (i) to designate the taxon as a nomen dubium and (ii) to synonymize the taxon with *My. longiceps*. Reisz *et al*. [11] elected to follow the second option, which has been subsequently adopted by other workers (e.g. [5,14]). However, as suggested by Reisz *et al*. [11], the pattern of dentition shared by *B. fortsillensis* and *My. longiceps* may in fact be plesiomorphic for synapsids and its presence thus predicted to broadly occur in Mycterosaurinae. Subsequent work on mycterosaurines supports this hypothesis. Reisz & Modesto [3] indicated the presence of alveoli at the posterior-most region of the cultriform process in *H. scholtzi* and both *Me. romeri* [12] and *Me. efremovi* (this study) preserve the same pattern as in *My. longiceps* and *B. fortsillensis*. The only discrepancy between *Me. efremovi* and *B. fortsillensis* is Vaughn's [20] description of the confluence of the ridges framing the posterior concavity as a 'shelf' [20, p. 46], whereas it appears as a shallow depression in our observations of *Me. efremovi*. However, Vaughn's illustrations do not clearly show a shelf-like structure, and the descriptive disparity probably represents nothing more than semantic differences. Thus, *B. fortsillensis* should be regarded as a nomen dubium, rather than as a junior synonym of *My. longiceps*, as it cannot be presently distinguished from either *My. longiceps* or *Me. efremovi*.

## 4.2. Phylogenetic results

Analysis of the matrix of Brocklehurst *et al*. [13] in PAUP* 4.0a165 [21] with the original taxon sampling (58 taxa) and the addition of the Richards Spur taxon, *Limnoscelis* as the outgroup, and removal of the three wildcard taxa (*Caseopsis agilis*, *Ctenorhachis jacksoni* and *Angelosaurus dolani*) recovered 108 most parsimonious trees (MPTs) with a length of 786 steps. The Richards Spur taxon is recovered as the sister taxon to *Me. romeri* in 100% of MPTs. This pair forms a polytomy with *H. scholtzi* and *E. longiceps* in 100% of MPTs. *Mycterosaurus* is then the sister taxon to the grouping of these four taxa in all MPTs. The taxon sample was pruned to focus on varanopid interrelationships and to improve computation time for testing node support, restricting the in-groups to varanopids, caseasaurs and ophiacodontids (39 taxa included in re-analysis). This produced 36 MPTs with a length of 499 steps and no changes to the overall tree topology (figure 6). Bootstrapping with 100 replicates led to modest support for the traditional Mycterosaurinae (65%) but poor support for all intrarelationships (a single polytomy) (figure 6). Support for the traditional Varanodontinae is strong (90%).

These results provide strong evidence to support a close relationship between *Me. romeri* and *Me. efremovi*. As noted above, the anatomical differences between *Me. romeri* and *Me. efremovi* are

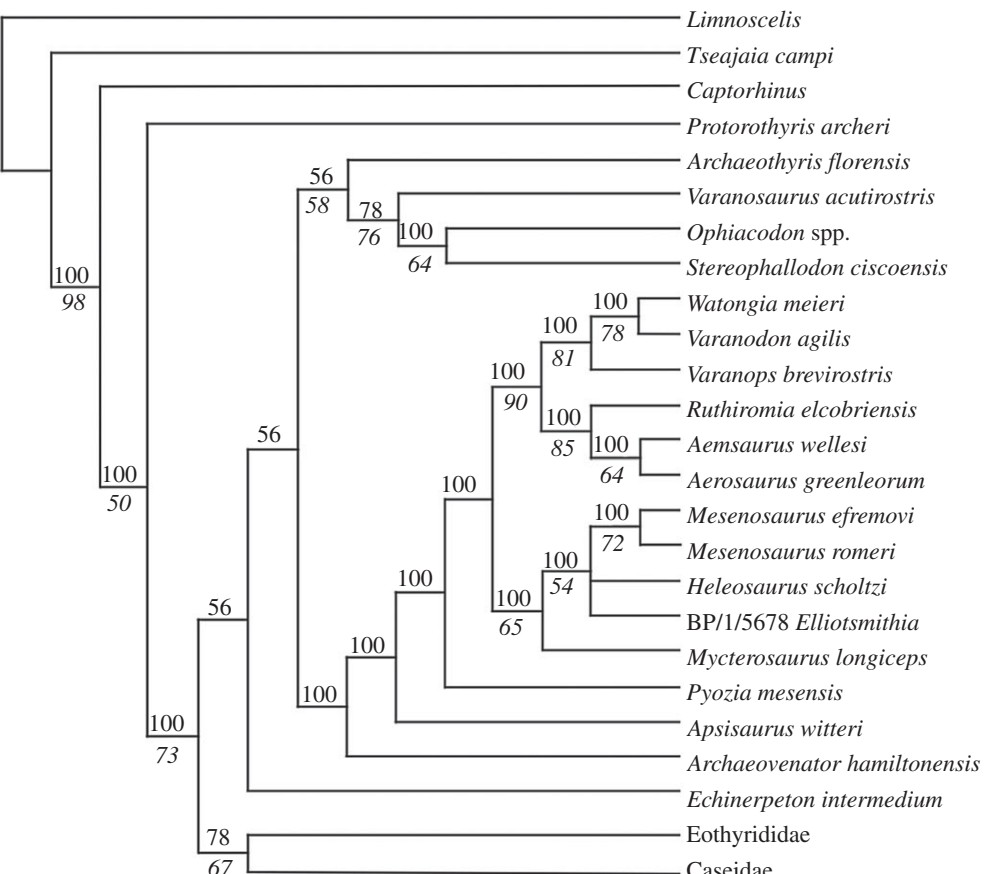

**Figure 6.** Majority-rule consensus tree showing the position of *Me. efremovi* sp. nov. in the matrix of Brocklehurst *et al.* [13]. Both Eothyrididae and Caseidae were recovered as monophyletic and are collapsed here for visual simplification of the outgroups. The nodal occurrence is indicated above the branch leading to a node and the bootstrap support value is indicated below the branch.

largely qualitative in nature, such as relative size and shape of the temporal fenestra, and are relatively minor, such as a slight difference in maxillary tooth count. At least some of the relatively minor qualitative differences could be attributed to ontogeny, given their correlation with size, since *Me. romeri* is smaller than *Me. efremovi*. Arguably, the most prominent distinction between these taxa is their temporal separation. Although the type locality of *Me. romeri* has not been absolutely dated, it is generally accepted to be middle to late Guadalupian in age (found in numerous localities in the Mezen River Basin in northern Russia, ranging in age from late Kazanian to late Tatarian, as indicated in Ivakhnenko [19]). The 289 Ma age (Artinskian) recovered for Richards Spur through radioisotopic dating of speleothems [22] thus produces a temporal gap exceeding 20 Myr. In general, this far exceeds the documented temporal range of most extinct tetrapods. However, it is important to emphasize that stratigraphic and temporal ranges are not diagnostic attributes, although they are still broadly informative. Making an *a priori* assumption that it is impossible or highly improbable for some genera to be unusually long-lived will result in a self-fulfilling prophecy of temporally restricted taxa. Examples of other Permo-Carboniferous tetrapods that persisted for exceptionally long temporal ranges do exist, such as the nectridean *Diplocaulus* [23], the temnospondyls *Aspidosaurus* [24], *Branchierpeton* [25] and *Eryops* [26], the eureptile *Captorhinus* [27], the diadectomorph *Diadectes* [28] and the sphenacodontid *Dimetrodon* [29]. A particularly important taxon in this regard is the pararareptile *Macroleter*, known from both Mezen and the Chickasha Formation of Oklahoma [30]. The age of the Chickasha Formation has long been disputed (see Olroyd & Sidor [4] for a summary), with the presence of *Macroleter* sometimes used as evidence for a Guadalupian age of the horizon [30]. However, without absolute dating of either the Chickasha Formation or the Mezen complex, it remains unclear whether these horizons are approximately coeval (or at least closer in age than suggested by Lucas [31]) or whether the observed distribution of *Macroleter* reflects a broad temporal range. The taxonomic longevity of varanopids (specifically *Mesenosaurus*) is less in question, given the combination of absolute dating for Richards Spur [22] in conjunction with marked differences in

faunal assemblages. This interpretation in turn suggests an ability of the clade to persist throughout different environments across Pangea and in the midst of pronounced faunal turnover throughout the Permian, and in particular, across the so-called Olson's gap. Thus, it is not entirely unexpected that some taxa could have been particularly successful and would have long temporal ranges.

An important taxon with respect to *Me. efremovi* is *C. trostheidei*, named for a headless skeleton from the Goldlauter Formation (Asselian/Sakmarian) of Germany [17]. This taxon (originally referred to as an indeterminate mesenosaurine from the Cabarz quarry) was recovered as the sister taxon to *Me. romeri* by Spindler *et al.* [15]. As frequent parallels have been made between early Permian deposits in south-central North America and the coeval Germanic deposits (in particular with the Bromacker quarry), it may be predicted that *C. trostheidei* and the new taxon, *Me. efremovi*, are closely related. However, it is impossible to test this at present because there is no skeletal overlap between these taxa (a reason why we elected not to test *Me. efremovi* in the matrix of Spindler *et al.* [15] or to test both *Me. efremovi* and *C. trostheidei* in the matrix of Brocklehurst *et al.* [13]). Extensive isolated varanopid postcranial remains have been collected at Richards Spur, but the presence of at least two other varanopids, *Mycterosaurus* and *Varanops* [5], complicates the identification of such remains without clear association with the more diagnostic cranial material. Future work will more thoroughly survey the large amount of isolated synapsid and varanopid postcranial material from Richards Spur to assess whether additional information can be incorporated into phylogenetic analyses. Spindler *et al.* [17] identified only a few qualitative features (captured in five phylogenetic characters) that differentiate *C. trostheidei* from *Me. romeri*. This is similar to our observations of a few (mutually exclusive at present) differences that differentiate *Me. efremovi* from *Me. romeri* (captured in two characters of Brocklehurst *et al.* [13]). At present, we do not consider the differences between *Me. romeri* and *Me. efremovi* to be sufficient to warrant separation at the genus level, and until either cranial material of *C. trostheidei* or confidently referable postcranial material of *Me. efremovi* are found, the relationship between these three taxa cannot be further evaluated in a phylogenetic framework.

## 4.3. Palaeogeographic implications

The presence and similarity of two species of *Mesenosaurus* separated by notable geographical (North America versus Russia) and temporal (Cisuralian versus Guadalupian) scales merits some discussion in order to contextualize this observation. The formation of Pangea during the Carboniferous allowed for a high degree of interconnectedness among geographically diverse tetrapod communities during the Permian. While many tetrapod clades exhibit taxonomic longevity across the Cisuralian and Guadalupian, it is accompanied by geographical shifts rather than static or cosmopolitan distributions. Taxonomic longevity across the Permian is indicative of either an ability to adapt to changes in ecology and climate [32] or remarkable environmental stability that facilitates the persistence of tetrapod communities. The latter is not well supported [33–35], and changes to palaeoenvironment are widely inferred to have exerted major influences on tetrapod evolution and community assemblages [36,37]. Most tetrapod clades exhibit a variety of morphological patterns across their temporal ranges. One of the most conspicuous patterns is changes in size. For example, dissorophid temnospondyls exhibit a major increase in size without major changes in morphology, with middle-late Permian taxa from Russia and China being at least twice as large as most early Permian taxa from North America [38,39]. Captorhinid eureptiles also exhibited a major increase in size over their evolution, although some relatively late-occurring taxa remained relatively small (e.g. [40]).

Size patterns of most synapsid clades are not directly comparable to other Permian tetrapods because of their relative taxonomic brevity. Synapsid diversity during the Cisuralian is characterized primarily by pelycosaurian-grade synapsids from palaeoequatorial localities of North America and Europe (e.g. [2,6,8,15,17]), whereas Guadalupian assemblages are characterized primarily by therapsids from Eurasia and South Africa [4]. Only two pelycosaurian clades extended through Olson's gap into the Guadalupian: caseids and varanopids. Whereas caseids remained relatively large-bodied, varanopids were progressively restricted to small-bodied taxa [3,22,32]. Furthermore, while both large- and small-bodied varanopids are found in the early Permian, middle Permian varanopids are mostly small-bodied taxa like *Mesenosaurus*, with the possible exception of *Watongia meieri*, if its post-Kungurian, middle Permian age is confirmed [41].

A compelling hypothesis for different size pattern evolution is diet. For example, caseids were large-bodied, high-fibre herbivores, similar to the coeval diadectids, which probably accounts for both their taxonomic longevity and lesser degree of size change [42,43]. Dissorophids were modestly sized faunivores in the early Permian, but the disappearance of other large-bodied early Permian

temnospondyls (e.g. eryopids, trematopids) may have allowed them to achieve greater body size, as may have occurred with *W. meieri* following the disappearance of the similarly faunivorous sphenacodontids [41]. The difference between these replacement scenarios may be attributed to the absence of any new clades of terrestrial temnospondyls that could have competed with dissorophids, whereas the radiation of therapsids may have produced greater competition for upper trophic niches among synapsids. A cautionary tale with respect to correlating size evolution and diet is the captorhinids. Although the clade became increasingly herbivorous throughout the Permian [16], there is no unidirectional shift towards large body size, and shifts in body size precede the evolution of high-fibre herbivory in this clade [40].

Varanopids were small- to medium-sized faunivores that remained relatively static in morphology throughout the Permian. The stark morphological similarity between *Me. efremovi* from the Cisuralian of North America and *Me. romeri* from the Guadalupian of Russia exemplifies this larger pattern within varanopids. Like dissorophids, varanopids were not the largest faunivores in most early Permian environments. However, this does not imply that they could not have been equally or more successful than taxa at higher trophic levels. We hypothesize that the evolutionary stasis of *Mesenosaurus* may be attributed to a conserved niche occupation throughout their temporal and geographical ranges. Given the relatively constrained size of varanopids and the restriction of later-occurring taxa to relatively small body sizes, it may be reasonably proposed that small-bodied varanopids like *Mesenosaurus* and *Mycterosaurus* occupied similar niches throughout their temporal ranges. As noted above, the radiation of therapsids may have limited the capacity for large-bodied varanopids to persist following the disappearance of other large pelycosaurian-grade synapsids (e.g. sphenacodontids). Thus, in spite of changing community assemblages, small-bodied varanopids may have successfully occupied similar niches within the trophic networks as relatively small faunivores with no competitive eco-equivalents present until the appearance of small diapsids near the end of the Permian [32]. This phenomenon implies an exceptionally high degree of extended ecological specialization of varanopids within terrestrial ecosystems that facilitated the persistence of *Mesenosaurus* across major faunal and environmental transitions of the Permian. Further research on the postcrania is required to assess if the same degree of stasis occurs throughout the entire skeleton and to explore the potential for specialization.

Ethics. All specimens described here are reposited in the Sam Noble Oklahoma Museum of Natural History (OMNH).
Data accessibility. The phylogenetic matrix used in this study has been uploaded as part of the electronic supplementary material. The electronic version of this article in portable document format will represent a published work according to the International Commission on Zoological Nomenclature (ICZN), and hence the new names contained in the electronic version are effectively published under that Code from the electronic edition alone. This published work and the nomenclatural acts it contains have been registered in ZooBank, the online registration system for the ICZN. The ZooBank Life Science Identifiers (LSIDs) can be resolved and the associated information viewed through any standard Web browser by appending the LSID to the prefix http://zoobank.org/. The LSID for this publication is: urn:lsid:zoobank.org:pub:F80E30F1-0425-4A5D-804E-45BFF542D86D.
Authors' contributions. R.R.R. conceived and designed the study; S.M. drafted the manuscript; B.M.G. performed the phylogenetic analysis and contributed to the manuscript; S.M., B.M.G. and R.R.R. edited the manuscript and approved it for final submission.
Competing interests. All authors declare no competing interests.
Funding. This work was supported by a Natural Sciences and Engineering Research Council (NSERC) Discovery Grant, and Jilin University to R.R.R.; an Ontario Graduate Scholarship (OGS) to B.M.G.
Acknowledgements. Thanks to Diane Scott for preparation and photography of the specimens and to Nicola Horsman for illustrations. Thanks to the staff of the Sam Noble Oklahoma Museum of Natural History for their continued support of our research on the Richards Spur locality. We also wish to acknowledge Bill May for his dedication to the collection of numerous specimens from this locality, and his generosity in sharing them with our research group. Thanks to the three anonymous reviewers for constructive comments that improved this manuscript.

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
