## [Reviewer comments · Royal Society Open Science]

Review History

RSOS-191297.R0 (Original submission)

Review form: Reviewer 1 (Christian F. Kammerer)

Is the manuscript scientifically sound in its present form?

Yes

Are the interpretations and conclusions justified by the results?

Yes

Is the language acceptable?

Yes

Do you have any ethical concerns with this paper?

No

Have you any concerns about statistical analyses in this paper?

No

Recommendation?

Accept with minor revision (please list in comments)

Comments to the Author(s)

See attached file (Appendix A).

Review form: Reviewer 2

Is the manuscript scientifically sound in its present form?

Yes

Are the interpretations and conclusions justified by the results?

Yes

Is the language acceptable?

Yes

Do you have any ethical concerns with this paper?

No

Have you any concerns about statistical analyses in this paper?

No

Recommendation?

Accept with minor revision (please list in comments)

Comments to the Author(s)

This submission will make a fantastic contribution to the paleontology of early amniotes. The text is well written, the interpretations are solid, and the illustrations are excellent. It will be a well-cited work for researchers of early synapsids or early reptiles (or both) and be of general interest to the other evolutionary biologists. I find the submission acceptable with very minor revisions, collated as follows:

line 12: typo in 'others'

line 47: semi-colon should be lightface

line 69: an individual gnathostome has a single mandible (composed of left and right rami)

line 153: change 'while' to 'whereas'

lines 162-163: the entrance of the lacrimal duct (lacrimal puncture or puncti) should be labelled (as some of our colleagues are using the lateral expression of this opening as a phylogenetic character)

line 229: what exactly is 'it' ?

line 244: delete the word 'on' near end of sentence

lines 246-257: any evidence for a tympanic flange ?

line 306: the angular generally articulates dorsally with the surangular

line 309: the surangular generally articulates ventrally with the angular; also, close up the space following the figure citation

line 324: change 'nares' to 'naris'

lines 360-361: do you mean that *Basicranodon fortsillensis* can be distinguished from neither *Mycterosaurus longiceps* nor *Mesenosaurus romeri* ?

lines 352, 354: '*M. longiceps*' should be '*My. longiceps*'

line 442: the term 'equatorial exposures' is somewhat misleading. Do you mean 'palaeoequatorial localities' ?

line 450: for 'explanator' do you mean 'hypothesis' ?

Review form: Reviewer 3 (David Ford)

Is the manuscript scientifically sound in its present form?

Yes

Are the interpretations and conclusions justified by the results?

Yes

Is the language acceptable?

Yes

Do you have any ethical concerns with this paper?

No

Have you any concerns about statistical analyses in this paper?

No

Recommendation?

Accept with minor revision (please list in comments)

Comments to the Author(s)

To the Editor/ Authors,

Thank you for the opportunity to review 'A new varanopid synapsids from the early Permian of Oklahoma and the evolutionary statuses of this clade' by Maho, Gee and Reisz. The authors have presented a descriptive study of new material from the prodigious Richards Spur locality, comprised of the three skulls and some elements of anterior cervical vertebrae, which the authors identify as belonging to a single varanopid. The authors performed a phylogenetic maximum parsimony analysis, using a previously published dataset, which recovers the new material as a sister taxon to *Mesenosaurus romeri*, from the middle Permian of Russia, and raised a new species, *Mesenosaurus efremovi*, to which the new material is allocated. They go on to discuss the implications of a relatively long-lived genus, within the context of varanopid longevity and

morphological status in general, in respect to niche conservation over wide temporal and spatial distances.

Varanopids are an important clade of early amniotes, and any new data, particularly such well-preserved specimens, is a thoroughly welcome addition to our growing understanding of varanopid evolution, particularly in light of alternative hypotheses of their affinities (Ford and Benson 2018). Consequently, I recommend publication of this study. However, there are a number of issues which I believe should be addressed by the authors prior to publication. Many are related to the descriptive passages on the manuscript, but others raise questions on the phylogenetic analysis. I detail these issues in the comments to authors

Diagnosis - line 64 - 'lack of contact between postorbital and supratemporal bones'. It would appear from the figures, and particularly from Fig 5, that the postorbital and supratemporal in *M. efremovi* are in contact, and that this contact distinguishes the new taxon from *M. romeri*, where these bones do not make contact (Reisz and Berman 2001). This view is indeed confirmed by the authors in lines 193-195. However, the same apparent error is made by the authors in the Discussion, lines 336-337. This requires clarification and amendment.

Skull roof - line 119 - is the premaxilla transversely (mediolaterally) narrow?

Skull roof - line 120 - are the authors able to note the nasal shelf is smoothly rounded between the medial and lateral surface off the premaxilla? This might be worth noting as it has been proposed as a synapomorphy of varanopids (Reisz and Berman 2001).

Maxilla - line 138/139 - does the dorsal process extend posterior to the dorsal process of the jugal? In 73500 it appears to do so in left dorsolateral view, but in right lateral view it appears to be level with the dorsal process of the jugal. In line 146 authors note that the suborbital process of the maxilla terminates at the posterior orbital margin, which is probably slightly anterior to the dorsal process of the jugal.

Maxilla - line 142 - 'The long dorsal process...' do the authors mean anteroposteriorly long, or dorsoventrally tall?

Maxilla - line 144 - 'proximity of the anterior orbital margin', perhaps authors could be more specific here, as in Reisz and Berman (2001), in describing the anterior extent of the lacrimal in respect to the snout, i.e. 40% between the orbit and the naris?

Maxilla - line 147 - is the anterior-most foramen larger than the posterior foramina? This might be of diagnostic interest in the wider sense.

Marginal dentition - line 155 - authors note in line 152 that there are four distinct, enlarged teeth in the caniniform region. In line 155 they note the region covers positions 4-6, i.e. only three teeth.

Lacrimal - line 164 - is the lacrimal excluded from the nasal bed contact of the maxilla with the nasal and the prefrontal?

Nasal - line 165 - should read 'with the premaxilla anteriorly..'

Nasal - line 166 - and with the prefrontal posterolaterally?

Prefrontal - line 175-177 - the prefrontal does articulate with the frontal medially, but not with the parietal or postfrontal or postorbital. I think there is some confusion here between the pre- and post-frontals.

Prefrontal - line 178 - this might be better described as a sharp right-angled transition between the lateral and dorsal surfaces of the prefrontal, as in other mycterosaurines.

Frontal - line 180 - '....in both species of Mesenosaurus' might be more explanatory.

Squamosal - is there a small posterodorsal process on the squamosal? The drawing of 73500 in right lateral seem to show a small process, which is interesting in respect to the more distinct processes on *Elliotsmithia* (Modesto et al. 2001). Without a close look at the specimen it's difficult to judge from the photo.

Vomer - line 237/238 - it's difficult to see in Fig 4 that the vomer, palatine and pterygoid articulate in the way described. In Fig.4, I think the labelling on the lower of the two pterygoids (pt) is incorrect, and this should be the palatine. Nevertheless, although the text describes a typical palatal layout, it's difficult to see this in the illustration.

Vomer - line 239m - it's interesting to see the anterior vomers are bifurcated. Are they formed asymmetrically in the same way as *M. romeri*, with the longer medial spur?

Pterygoid - line 247 - does the palatal rams of the pterygoid articulate with the ectopterygoid anterolaterally or posterolaterally?

Pterygoid - line 254/255 - are the teeth on the transverse flange isodont, or is the posterior-most row composed of larger teeth?

Pterygoid - line 256/257 - Fig 4 doesn't show any contact between the quadrate ramus of the pterygoid and the quadrate. It just appears to end at the squamosal. In all it contacts the anterior flange of the quadrate, but this can't be seen in the figure.

Exoccipital - line 275- the exoccipitals are paired elements but rarely meet their partner along the midline since they are often separated by the basioccipital (ventrally) and by the supraoccipital (dorsally) as in the case of *M. Romeri* (see Reisz and Berman Fig 4). Do the authors mean that each exoccipital is formed of two elements sutured along the sagittal midline?

Stapes - line 298 - I'm not sure what the authors mean by the stapedia foramen piercing the columella 'anteroposteriorly to the footplate'. Perhaps this could be expanded upon.

Dentary - line 293 - the dentary forms the anterodorsal margin of the mandible.

Coronoid - line 304 - '...frame the coronoid process from within', I suspect that authors mean that the coronoid has a mainly medial (lingual) exposure.

Angular - line 306m- doesn't the angular articulate with the surangular dorsally (Fig 5) rather than anteriorly?

Surangular - line 309 - Fig 5 seems to show the surangular articulating with the dentary anterodorsally and the angular ventrally.

Discussion - line 329 - are the authors sure that premaxillary teeth are absent in other mycterosaurines? Or is it that other mycterosaurines have less than 5 premaxillary teeth?

Discussion - line 360 - the authors note that the parasphenoid described by Vaughn (1958) cannot be distinguished from *M. efremovi* and *Mycterosaurus*, and should be considered a nomen dubium. This may well be the case, however, the median ventral ridge on the basal plate described by the

authors (lines 269/270) does not appear to be present in the parasphenoid described by Vaughn, where the anterolateral ridges (crista ventrolaterales?) form a shelf at their anterior point of confluence rather than a medial ridge (line 270). Vaughn describes medial ridges posterior to this shelf, in the concavity between the posteriorly diverging crista ventrolaterales. Perhaps the authors could consider a close-up drawing or illustration of the parasphenoid. It would be very useful for comparative purposes with other early amniotes.

Phylogenetic results - line 370 - MPTS should be MPTs

Phylogenetic results - line 380 - is the difference in tooth count qualitative in nature, or quantitative?

Phylogenetic results - line 420 - did the authors consider adding Cabarzia to the Brocklehurst et al. matrix? Although only postcranial material is available for this taxon, postcranial data is available for M. Romeri (same paper, Spindler et al. 2018).

References:

Ford, D.P. and Benson, R.B., 2019. A redescription of *Orovenator mayorum* (Sauropsida, Diapsida) using high-resolution μ CT, and the consequences for early amniote phylogeny. *Papers in Palaeontology*, 5(2), pp.197-239.

Reisz, R.R. and Berman, D.S., 2001. The skull of *Mesenosaurus romeri*, a small varanopseid (Synapsida: Eupelycosauria) from the Upper Permian of the Mezen River Basin, northern Russia. *ANNALS-CARNEGIE MUSEUM PITTSBURGH*, 70(2), pp.113-132.

Modesto, S., Sidor, C.A., Rubidge, B.S. and Welman, J., 2001. A second varanopseid skull from the Upper Permian of South Africa: implications for Late Permian 'pelycosaur' evolution. *Lethaia*, 34(4), pp.249-259.

Spindler, F., Werneburg, R. and Schneider, J.W., 2019. A new mesenosaurine from the lower Permian of Germany and the postcrania of *Mesenosaurus*: implications for early amniote comparative osteology. *PalZ*, pp.1-42.

Decision letter (RSOS-191297.R0)

03-Sep-2019

Dear Dr Maho

On behalf of the Editors, I am pleased to inform you that your Manuscript RSOS-191297 entitled "A new varanopid synapsid from the early Permian of Oklahoma and the evolutionary stasis in this clade" has been accepted for publication in Royal Society Open Science subject to minor revision in accordance with the referee suggestions. Please find the referees' comments at the end of this email.

The reviewers and handling editors have recommended publication, but also suggest some minor

revisions to your manuscript. Therefore, I invite you to respond to the comments and revise your manuscript.

- Ethics statement

- Data accessibility

If you wish to submit your supporting data or code to Dryad (<http://datadryad.org/>), or modify your current submission to dryad, please use the following link:
<http://datadryad.org/submit?journalID=RSOS&manu=RSOS-191297>

- Competing interests

- Authors' contributions

- Acknowledgements

- Funding statement

Please ensure you have prepared your revision in accordance with the guidance at

<https://royalsociety.org/journals/authors/author-guidelines/> -- please note that we cannot publish your manuscript without the end statements. We have included a screenshot example of the end statements for reference. If you feel that a given heading is not relevant to your paper, please nevertheless include the heading and explicitly state that it is not relevant to your work.

Because the schedule for publication is very tight, it is a condition of publication that you submit the revised version of your manuscript before 12-Sep-2019. Please note that the revision deadline will expire at 00.00am on this date. If you do not think you will be able to meet this date please let me know immediately.

Please note that Royal Society Open Science charge article processing charges for all new

submissions that are accepted for publication. Charges will also apply to papers transferred to Royal Society Open Science from other Royal Society Publishing journals, as well as papers submitted as part of our collaboration with the Royal Society of Chemistry (<http://rsos.royalsocietypublishing.org/chemistry>).

on behalf of Dr Julia Brenda Desojo (Associate Editor) and Kevin Padian (Subject Editor)
openscience@royalsociety.org

Reviewer comments to Author:
Reviewer: 1

Comments to the Author(s)
See attached file.

Reviewer: 2

Comments to the Author(s)
This submission will make a fantastic contribution to the paleontology of early amniotes. The text is well written, the interpretations are solid, and the illustrations are excellent. It will be a well-cited work for researchers of early synapsids or early reptiles (or both) and be of general interest to the other evolutionary biologists. I find the submission acceptable with very minor revisions, collated as follows:

line 12: typo in 'others'

line 47: semi-colon should be lightface

line 69: an individual gnathostome has a single mandible (composed of left and right rami)

line 153: change 'while' to 'whereas'

lines 162-163: the entrance of the lacrimal duct (lacrimal puncture or puncti) should be labelled (as some of our colleagues are using the lateral expression of this opening as a phylogenetic character)

line 229: what exactly is 'it' ?

line 244: delete the word 'on' near end of sentence

lines 246-257: any evidence for a tympanic flange ?

line 306: the angular generally articulates dorsally with the surangular

line 309: the surangular generally articulates ventrally with the angular; also, close up the space following the figure citation

line 324: change 'nares' to 'naris'

lines 360-361: do you mean that *Basicranodon fortsillensis* can be distinguished from neither *Mycterosaurus longiceps* nor *Mesenosaurus romeri* ?

lines 352, 354: '*M. longiceps*' should be '*My. longiceps*'

line 442: the term 'equatorial exposures' is somewhat misleading. Do you mean 'palaeoequatorial localities' ?

line 450: for 'explanator' do you mean 'hypothesis' ?

Reviewer: 3

Comments to the Author(s)

To the Editor/ Authors,

Thank you for the opportunity to review 'A new varanopid synapsids from the early Permian of Oklahoma and the evolutionary statuses of this clade' by Maho, Gee and Reisz. The authors have presented a descriptive study of new material from the prodigious Richards Spur locality, comprised of the three skulls and some elements of anterior cervical vertebrae, which the authors identify as belonging to a single varanopid. The authors performed a phylogenetic maximum parsimony analysis, using a previously published dataset, which recovers the new material as a sister taxon to *Mesenosaurus romeri*, from the middle Permian of Russia, and raised a new species, *Mesenosaurus efremovi*, to which the new material is allocated. They go on to discuss the implications of a relatively long-lived genus, within the context of varanopid longevity and morphological stasis in general, in respect to niche conservation over wide temporal and spatial distances.

Varanopids are an important clade of early amniotes, and any new data, particularly such well-preserved specimens, is a thoroughly welcome addition to our growing understanding of varanopid evolution, particularly in light of alternative hypotheses of their affinities (Ford and Benson 2018). Consequently, I recommend publication of this study. However, there are a number of issues which I believe should be addressed by the authors prior to publication. Many are related to the descriptive passages on the manuscript, but others raise questions on the phylogenetic analysis. I detail these issues in the comments to authors

Diagnosis - line 64 - 'lack of contact between postorbital and supratemporal bones'. It would appear from the figures, and particularly from Fig 5, that the postorbital and supratemporal in

M. efremovi are in contact, and that this contact distinguishes the new taxon from M. romeri, where these bones do not make contact (Reisz and Berman 2001). This view is indeed confirmed by the authors in lines 193-195. However, the same apparent error is made by the authors in the Discussion, lines 336-337. This requires clarification and amendment.

Skull roof - line 119 - is the premaxilla transversely (mediolaterally) narrow?

Skull roof - line 120 - are the authors able to note the nasal shelf is smoothly rounded between the medial and lateral surface of the premaxilla? This might be worth noting as it has been proposed as a synapomorphy of varanopids (Reisz and Berman 2001).

Maxilla - line 138/139 - does the dorsal process extend posterior to the dorsal process of the jugal? In 73500 it appears to do so in left dorsolateral view, but in right lateral view it appears to be level with the dorsal process of the jugal. In line 146 authors note that the suborbital process of the maxilla terminates at the posterior orbital margin, which is probably slightly anterior to the dorsal process of the jugal.

Maxilla - line 142 - 'The long dorsal process...' do the authors mean anteroposteriorly long, or dorsoventrally tall?

Maxilla - line 144 - 'proximity of the anterior orbital margin', perhaps authors could be more specific here, as in Reisz and Berman (2001), in describing the anterior extent of the lacrimal in respect to the snout, i.e. 40% between the orbit and the naris?

Maxilla - line 147 - is the anterior-most foramen larger than the posterior foramina? This might be of diagnostic interest in the wider sense.

Marginal dentition - line 155 - authors note in line 152 that there are four distinct, enlarged teeth in the caniniform region. In line 155 they note the region covers positions 4-6, i.e. only three teeth.

Lacrimal - line 164 - is the lacrimal excluded from the naris bed contact of the maxilla with the nasal and the prefrontal?

Nasal - line 165 - should read 'with the premaxilla anteriorly..'

Nasal - line 166 - and with the prefrontal posterolaterally?

Prefrontal - line 175-177 - the prefrontal does articulate with the frontal medially, but not with the parietal or postfrontal or postorbital. I think there is some confusion here between the pre- and post-frontals.

Prefrontal - line 178 - this might be better described as a sharp right-angled transition between the lateral and dorsal surfaces of the prefrontal, as in other mycterosaurines.

Frontal - line 180 - '....in both species of Mesenosaurus' might be more explanatory.

Squamosal - is there a small posterodorsal process on the squamosal? The drawing of 73500 in right lateral view seem to show a small process, which is interesting in respect to the more distinct processes on *Elliotsmithia* (Modesto et al. 2001). Without a close look at the specimen it's difficult to judge from the photo.

Vomer - line 237/238 - it's difficult to see in Fig 4 that the vomer, palatine and pterygoid articulate in the way described. In Fig.4, I think the labelling on the lower of the two pterygoids (pt) in

incorrect, and this should be the palatine. Nevertheless, although the text describes a typical palatal layout, it's difficult to see this in the illustration.

Vomer - line 239m - it's interesting to see the anterior vomers are bifurcated. Are they formed asymmetrically in the same way as *M. romeri*, with the longer medial spur?

Pterygoid - line 247 - do the palatal rami of the pterygoid articulate with the ectopterygoid anterolaterally or posterolaterally?

Pterygoid - line 254/255 - are the teeth on the transverse flange isodont, or is the posterior-most row composed of larger teeth?

Pterygoid - line 256/257 - Fig 4 doesn't show any contact between the quadrate ramus of the pterygoid and the quadrate. It just appears to end at the squamosal. In all it contacts the anterior flange of the quadrate, but this can't be seen in the figure.

Exoccipital - line 275 - the exoccipitals are paired elements but rarely meet their partner along the midline since they are often separated by the basioccipital (ventrally) and by the supraoccipital (dorsally) as in the case of *M. Romeri* (see Reisz and Berman Fig 4). Do the authors mean that each exoccipital is formed of two elements sutured along the sagittal midline?

Stapes - line 298 - I'm not sure what the authors mean by the stapedial foramen piercing the columella 'anteroposteriorly to the footplate'. Perhaps this could be expanded upon.

Dentary - line 293 - the dentary forms the anterodorsal margin of the mandible.

Coronoid - line 304 - '...frame the coronoid process from within', I suspect that authors mean that the coronoid has a mainly medial (lingual) exposure.

Angular - line 306m - doesn't the angular articulate with the surangular dorsally (Fig 5) rather than anteriorly?

Surangular - line 309 - Fig 5 seems to show the surangular articulating with the dentary anterodorsally and the angular ventrally.

Discussion - line 329 - are the authors sure that premaxillary teeth are absent in other mycterosaurines? Or is it that other mycterosaurines have less than 5 premaxillary teeth?

Discussion - line 360 - the authors note that the parasphenoid described by Vaughn (1958) cannot be distinguished from *M. efremovi* and *Mycterosaurus*, and should be considered a nomen dubium. This may well be the case, however, the median ventral ridge on the basal plate described by the authors (lines 269/270) does not appear to be present in the parasphenoid described by Vaughn, where the anterolateral ridges (crista ventrolaterales?) form a shelf at their anterior point of confluence rather than a medial ridge (line 270). Vaughn describes medial ridges posterior to this shelf, in the concavity between the posteriorly diverging crista ventrolaterales. Perhaps the authors could consider a close-up drawing or illustration of the parasphenoid. It would be very useful for comparative purposes with other early amniotes.

Phylogenetic results - line 370 - MPTS should be MPTs

Phylogenetic results - line 380 - is the difference in tooth count qualitative in nature, or quantitative?

Phylogenetic results - line 420 - did the authors consider adding Cabarzia to the Brocklehurst et al. matrix? Although only postcranial material is available for this taxon, postcranial data is available for *M. Romeri* (same paper, Spindler et al. 2018).

References:

Ford, D.P. and Benson, R.B., 2019. A redescription of *Orovenator mayorum* (Sauropsida, Diapsida) using high-resolution μ CT, and the consequences for early amniote phylogeny. *Papers in Palaeontology*, 5(2), pp.197-239.

Reisz, R.R. and Berman, D.S., 2001. The skull of *Mesenosaurus romeri*, a small varanopseid (Synapsida: Eupelycosauria) from the Upper Permian of the Mezen River Basin, northern Russia. *ANNALS-CARNEGIE MUSEUM PITTSBURGH*, 70(2), pp.113-132.

Modesto, S., Sidor, C.A., Rubidge, B.S. and Welman, J., 2001. A second varanopseid skull from the Upper Permian of South Africa: implications for Late Permian 'pelycosaur' evolution. *Lethaia*, 34(4), pp.249-259.

Spindler, F., Werneburg, R. and Schneider, J.W., 2019. A new mesenosaurine from the lower Permian of Germany and the postcrania of *Mesenosaurus*: implications for early amniote comparative osteology. *PalZ*, pp.1-42.

Author's Response to Decision Letter for (RSOS-191297.R0)

See Appendix B.

Decision letter (RSOS-191297.R1)

24-Sep-2019

Dear Ms Maho,

I am pleased to inform you that your manuscript entitled "A new varanopid synapsid from the early Permian of Oklahoma and the evolutionary stasis in this clade" is now accepted for publication in Royal Society Open Science.

Kind regards,

Lianne Parkhouse
Royal Society Open Science
openscience@royalsociety.org

on behalf of Dr Julia Brenda Desojo (Associate Editor) and Kevin Padian (Subject Editor)
openscience@royalsociety.org

Appendix A

These are beautiful and anatomically very informative specimens, which are well described and figured by the authors. In general this is an important contribution to our knowledge of varanopid anatomy and diversity in American Permian faunas.

Regarding the argument that these fossils indicate remarkable morphological stasis in a Permian tetrapod, I think the authors are probably correct that *Mesenosaurus* was long ranging. However, this argument is rendered highly uncertain by the essentially unknown date of the Russian Mezen assemblage. Historically considered upper Permian due to the presence of therapsids and mostly considered middle Permian now, this assemblage is dominated by parareptiles and pelycosaurs and has a very archaic aspect. The therapsid components are also exceedingly plesiomorphic, probably near the divergence of the major therapsid groups. I would not be surprised if the Mezen fauna was substantially older than usually thought, potentially filling in Olson's Gap to some degree. With that said, a date as old as Artinskian (as for Richards Spur) is unlikely, and *Mesenosaurus* likely did range through several stages of the Permian. Relevant to this topic and unremarked upon by the authors (although the paper is cited) is the cross-continental presence of the nycteroleterid tetrapod *Macroleter*, previously known from the Mezen River and later recognized in seemingly earlier Permian strata in Oklahoma. This is an intriguingly comparable scenario to the present discovery of *Mesenosaurus* and warrants discussion, as it suggests broader spatial distribution of Mezen-style taxa (and again raises the issue of how much temporal separation there really is between these faunas).

Although a historically-used variant and acceptable transliteration, "Mesen" is usually spelled "Mezen" in Latin script (Google search recovers <200 hits for "Mesen" vs. nearly 8000 for "Mezen") and this should be followed here.

Minor edits:

Line 3: change "extending from" to "whose range extends from"

Line 8: change "from fragmentary disarticulated material" to "from fragmentary disarticulated material at Richards Spur" (as these taxa are known from complete skulls at other localities)

Line 62: change "that do not extend" to "that does not extend"

Line 102: "are partially represented to complete" is somewhat awkward, I recommend changing this to "range from partial to complete"

Lines 244–245: change "exposing only on the dorsal surfaces" to "with only the dorsal surfaces exposed"

Line 278: change "to from a" to "to form a"

Line 312: delete the extraneous "on the"

Lines 328–329: add commas after "features" and "positions"

Line 333: change "relatively short" to "is relatively short"

Lines 344–345: change "certainly not sufficient to argue for taxonomic distinction above the species level" (although I agree this is the case) to "and we would argue insufficient for taxonomic distinction above the species level"

Appendix B

Response to Reviewers

We thank the three reviewers and the editor for their constructive feedback on this manuscript. Below are our point-by-point responses to the comments raised during review.

Reviewer: 1

- Regarding the argument that these fossils indicate remarkable morphological stasis in a Permian tetrapod, I think the authors are probably correct that Mesenosaurus was long ranging. However, this argument is rendered highly uncertain by the essentially unknown date of the Russian Mezen assemblage. Historically considered upper Permian due to the presence of therapsids and mostly considered middle Permian now, this assemblage is dominated by parareptiles and pelycosaurids and has a very archaic aspect. The therapsid components are also exceedingly plesiomorphic, probably near the divergence of the major therapsid groups. I would not be surprised if the Mezen fauna was substantially older than usually thought, potentially filling in Olson's Gap to some degree. With that said, a date as old as Artinskian (as for Richards Spur) is unlikely, and Mesenosaurus likely did range through several stages of the Permian. Relevant to this topic and unremarked upon by the authors (although the paper is cited) is the cross-continental presence of the nycteroleterid tetrapod Macroleter, previously known from the Mezen River and later recognized in seemingly earlier Permian strata in Oklahoma. This is an intriguingly comparable scenario to the present discovery of Mesenosaurus and warrants discussion, as it suggests broader spatial distribution of Mezen-style taxa (and again raises the issue of how much temporal separation there really is between these faunas).
 - We added a few brief comments on this in the discussion.
- Although a historically-used variant and acceptable transliteration, "Mesen" is usually spelled "Mezen" in Latin script (Google search recovers <200 hits for "Mesen" vs. nearly 8000 for "Mezen") and this should be followed here.
 - We made the spelling modification throughout the manuscript.
- Line 3: change "extending from" to "whose range extends from"
 - We made the requested edit.
- Line 8: change "from fragmentary disarticulated material" to "from fragmentary disarticulated material at Richards Spur" (as these taxa are known from complete skulls at other localities)
 - We made the requested edit.
- Line 62: change "that do not extend" to "that does not extend"
 - We made the requested edit.
- Line 102: "are partially represented to complete" is somewhat awkward, I recommend changing this to "range from partial to complete"
 - We made the requested edit.
- Lines 244–245: change "exposing only on the dorsal surfaces" to "with only the dorsal surfaces exposed"

Response to Reviewers

- We made the requested edit.
- Line 278: change “to from a” to “to form a”
 - We made the requested edit.
- Line 312: delete the extraneous “on the”
 - We made the edit.
- Lines 328–329: add commas after “features” and “positions”
 - We added a coma after “positions” but don't believe that one is grammatically warranted after “features.”
- Line 333: change “relatively short” to “is relatively short”
 - We made the requested edit.
- Lines 344–345: change “certainly not sufficient to argue for taxonomic distinction above the species level” (although I agree this is the case) to “and we would argue insufficient for taxonomic distinction above the species level”
 - We made the requested edit.

Reviewer: 2

- line 12: typo in ‘others’
 - We made the correction.
- line 47: semi-colon should be lightface
 - We made the correction.
- line 69: an individual gnathostome has a single mandible (composed of left and right rami)
 - We made the correction.
- line 153: change ‘while’ to ‘whereas’
 - We made the requested edit.
- lines 162-163: the entrance of the lacrimal duct (lacrimal puncture or puncti) should be labelled (as some of our colleagues are using the lateral expression of this opening as a phylogenetic character)
 - We added the additional label to the figures.
- line 229: what exactly is ‘it’ ?
 - We clarified the language here.
- line 244: delete the word ‘on’ near end of sentence

Response to Reviewers

- We made the correction.
- lines 246-257: any evidence for a tympanic flange?
 - We added additional comments to this end.
- line 306: the angular generally articulates dorsally with the surangular
 - We made the correction.
- line 309: the surangular generally articulates ventrally with the angular; also, close up the space following the figure citation
 - We made the corrections.
- line 324: change ‘nares’ to ‘naris’
 - We made the requested edit.
- lines 360-361: do you mean that *Basicranodon fortsillensis* can be distinguished from neither *Mycterosaurus longiceps* nor *Mesenosaurus romeri* ?
 - We clarified the language here.
- lines 352, 354: ‘*M. longiceps*’ should be ‘*My. longiceps*’
 - We made the requested edit.
- line 442: the term ‘equatorial exposures’ is somewhat misleading. Do you mean ‘palaeoequatorial localities’?
 - We made the requested edit.
- line 450: for ‘explanator’ do you mean ‘hypothesis’?
 - We made the requested edit.

Reviewer: 3

- Diagnosis - line 64 - ‘lack of contact between postorbital and supratemporal bones’. It would appear from the figures, and particularly from Fig 5, that the postorbital and supratemporal in *M. efremovi* are in contact, and that this contact distinguishes the new taxon from *M. romeri*, where these bones do not make contact (Reisz and Berman 2001). This view is indeed confirmed by the authors in lines 193-195. However, the same apparent error is made by the authors in the Discussion, lines 336-337. This requires clarification and amendment.
 - We clarified the language throughout the manuscript.
- Skull roof - line 119 - is the premaxilla transversely (mediolaterally) narrow?
 - We clarified the details.
- Skull roof - line 120 - are the authors able to note the narial shelf is smoothly rounded between the medial and lateral surface off the premaxilla? This might be worth noting as it has been proposed as a synapomorphy of varanopids (Reisz and Berman 2001).

Response to Reviewers

- We added additional details.
- Maxilla - line 138/139 - does the dorsal process extend posterior to the dorsal process of the jugal? In 73500 it appears to do so I left dorsolateral view, but in right lateral view it appears to be level with the dorsal process of the jugal. In line 146 authors note that the suborbital process of the maxilla terminates at the posterior orbital margin, which is probably slightly anterior to the dorsal process of the jugal.
 - We clarified the language.
- Maxilla - line 144 - 'proximity of the anterior orbital margin', perhaps authors could be more specific here, as in Reisz and Berman (2001), in describing the anterior extent of the lacrimal in respect to the snout, i.e. 40% between the orbit and the naris?
 - We clarified the language and added additional details.
- Maxilla- line 147 - is the anterior-most foramen larger than the posterior foramina? This might be of diagnostic interest in the wider sense.
 - We clarified the language.
- Marginal dentition - line 155 - authors note in line 152 that there are four distinct, enlarged teeth in the caniniform region. In line 155 they note the region covers positions 4-6, i.e only three teeth.
 - We clarified the language.
- Lacrimal - line 164 - is the lacrimal excluded from the naris bed contact of the maxilla with the nasal and the prefrontal?
 - We clarified the language.
- Nasal - line 165 - should read 'with the premaxilla anteriorly..'
 - We made the requested edit.
- Nasal - line 166 - and with the prefrontal posterolaterally?
 - We made the edit.
- Prefrontal - line 175-177 - the prefrontal does articulate with the frontal medially, but not with the parietal or postfrontal or postorbital. I think there is some confusion here between the pre- and post-frontals.
 - We made the correction.
- Prefrontal - line 178 - this might be better described as a sharp right-angled transition between the lateral and dorsal surfaces of the prefrontal, as in other mycterosaurines.
 - We made the requested edit.
- Frontal - line 180 - '....in both species of Mesenosaurus' might be more explanatory.
 - We made the requested edit.

Response to Reviewers

- Squamosal - is there a small posterodorsal process on the squamosal? The drawing of 73500 in right lateral seem to show a small process, which is interesting in respect to the more distinct processes on *Elliotsmithia* (Modesto et al. 2001). Without a close look at the specimen it's difficult to judge from the photo.
 - We added additional details.
- Vomer - line 237/238 - it's difficult to see in Fig 4 that the vomer, palatine and pterygoid articular in the way described. In Fig.4, I think the labelling on the lower of the two pterygoids (pt) is incorrect, and this should be the palatine. Nevertheless, although the text describes a typical palatal layout, it's difficult to see this in the illustration.
 - We made the changes to the description regarding sutural contacts but have double-checked the identifications and believe the figure was labeled correctly. The figured profile appears to show two separate ossifications, but they are exposed as a single ossification in views that were difficult to photograph.
- Vomer - line 239m - it's interesting to see the anterior vomers are bifurcated. Are they formed asymmetrically in the same way as *M. romeri*, with the longer medial spur?
 - We added additional information.
- Pterygoid - line 247 - does the palatal rami of the pterygoid articulate with the ectopterygoid anterolaterally or posterolaterally?
 - We checked the original interpretation and verified that it was anterolateral.
- Pterygoid - line 254/255 - are the teeth on the transverse flange isodont, or is the posterior-most row composed of larger teeth?
 - It was not possible to assess size patterns for this region in any specimen.
- Pterygoid - line 256/257 - Fig 4 doesn't show any contact between the quadrate ramus of the pterygoid and the quadrate. It just appears to end at the squamosal. In all it contacts the anterior flange of the quadrate, but this can't be seen in the figure.
 - We clarified the language.
- Exoccipital - line 275- the exoccipitals are paired elements but rarely meet their partner along the midline since they are often separated by the basioccipital (ventrally) and by the supraoccipital (dorsally) as in the case of *M. Romeri* (see Reisz and Berman Fig 4). Do the authors mean that each exoccipital is formed of two elements sutured along the sagittal midline?
 - We clarified the language.
- Stapes - line 298 - I'm not sure what the authors mean by the stapedial foramen piercing the columella 'anteroposteriorly to the footplate'. Perhaps this could be expanded upon.
 - We clarified the language.
- Dentary - line 293 - the dentary forms the anterodorsal margin of the mandible.
 - We made the edit.

Response to Reviewers

- Coronoid - line 304 - ‘...frame the coronoid process form within’, I suspect that authors mean that the coronoid has a mainly medial (lingual) exposure.
 - We clarified the language.
- Angular - line 306m- doesn’t the angular articulate with the surangular dorsally (Fig 5) rather than anteriorly?
 - We made the correction
- Surangular - line 309 - Fig 5 seems to show the surangular articulating with the dentary anterodorsally and the angular ventrally.
 - We made the corrections.
- Discussion - line 329 - are the authors sure that premaxillary teeth are absent in other mycterosaurines? Or is it that other mycterosaurines have less than 5 premaxillary teeth?
 - We clarified the language.
- Discussion - line 360 - the authors note that the parasphenoid described by Vaughn (1958) cannot be distinguish from *M. efremovi* and *Mycterosaurus*, and should be considered a nomen dubium. This may well be the case, however, the median ventral ridge on the basal plate described by the authors (lines 269/270) does not appear to be present in the parasphenoid described by Vaughn, where the anterolateral ridges (crista ventrolaterales?) form a shelf at their anterior point of confluence rather than a medial ridge (line 270). Vaughn describes medial ridges posterior to this shelf, in the concavity between the posteriorly diverging crista ventrolaterales. Perhaps the authors could consider a close-up drawing or illustration of the parasphenoid. It would be very useful for comparative purposes with other early amniotes.
 - We have clarified some of the details of the description of our taxon; there is a medial ridge in the concavity, but it is largely obscured by dislodged elements. A comparison with Vaughn’s figures did not indicate any appreciable difference in the anterior confluence of the ridges (i.e. no distinct shelf), and we attribute the apparent difference to word choice. We are working on CT data of the parasphenoid of the specimen that will be published in the future that will more clearly illustrate some of the relevant features in all anatomical profiles.
- Phylogenetic results - line 370 - MPTS should be MPTs
 - We made the edit.
- Phylogenetic results - line 380 - is the difference in tooth count qualitative in nature, or quantitative?
 - We clarified the language.
- Phylogenetic results - line 420 - did the authors consider adding *Cabarzia* to the Brocklehurst et al. matrix? Although only postcranial material is available for this taxon, postcranial data is available for *M. Romeri* (same paper, Spindler et al. 2018).
 - We did consider this option, but this would still fail to fully test the relationship of the three taxa, as the issue is the lack of skeletal overlap between the new taxon

Response to Reviewers

that we describe here and *Cabarzia*. Adding *Cabarzia* to the matrix that we used would essentially provide a single-taxon case study for partitioned datasets (i.e. does postcranial or cranial anatomy exert more influence on this specific relationship), which is something already tested by Benson (2012), which was the framework for the Brocklehurst et al. matrix.